# TabRepo: A Large Scale Repository of Tabular Model Evaluations and its AutoML Applications

**David Salinas**[1,*]  **Nick Erickson**[2,*]

[1]University of Freiburg
[2]Amazon Web Services
[*]Equal contribution

**Abstract**  We introduce TabRepo, a new dataset of tabular model evaluations and predictions. TabRepo contains the predictions and metrics of 1310 models evaluated on 200 classification and regression datasets. We illustrate the benefit of our dataset in multiple ways. First, we show that it allows to perform analysis such as comparing Hyperparameter Optimization against current AutoML systems while also considering ensembling at marginal cost by using precomputed model predictions. Second, we show that our dataset can be readily leveraged to perform transfer-learning. In particular, we show that applying standard transfer-learning techniques allows to outperform current state-of-the-art tabular systems in accuracy, runtime and latency.

## 1 Introduction

Machine learning on structured tabular data has a long history due to its wide range of practical applications. Significant progress has been achieved through improving supervised learning models, with key method landmarks including SVM (Hearst et al., 1998), Random Forest (Breiman, 2001) and Gradient Boosted Trees (Friedman, 2001). Recent methods include pretraining transformer models (Hollmann et al., 2022; Zhu et al., 2023) and combining non-parametric and deep-learning techniques (Gorishniy et al., 2023).

AutoML methods build upon base models to achieve superior performance and ease of use for ML practitioners (Thornton et al., 2013). Auto-Sklearn (Feurer et al., 2015, 2022) was an early approach that proposed to select pipelines to ensemble from the Sklearn library and meta-learn the hyperparameter optimization (HPO) with offline evaluations. The approach was successful and won several AutoML competitions. Several frameworks followed with other AutoML approaches such as TPOT (Olson and Moore, 2019), H2O AutoML (LeDell and Poirier, 2020), and AutoGluon (Erickson et al., 2020). AutoGluon showed particularly strong performance by combining bagging (Breiman, 1996), ensembling (Caruana et al., 2004), and multi-layer stacking (Wolpert, 1992).

The proliferation of supervised learning models and AutoML systems led to several works focusing on benchmarking tabular methods. Recently, Gijsbers et al. (2024) proposed the AutoML-Benchmark (AMLB), a unified benchmark to compare tabular methods. Because AMLB evaluates across 1040 tasks, evaluating a single method on AMLB requires 40000 CPU hours of compute[1]. Due to this cost, the benchmark is only run once a year, with heavy consideration taken to which methods should be included to mitigate compute concerns. This makes it expensive to perform thorough ablations and explore research directions. For instance, measuring the impact of ensembling requires running the ablated method on the entire benchmark for every setting tested.

To address this issue, we introduce TabRepo[2], a dataset of model evaluations and predictions. The main contributions of this paper are:

---

[1]The CPU hour requirement is based on running the full 104 datasets in AMLB across 10 folds for both 1 hour and 4 hour time limits on an 8 CPU machine.

[2]The dataset and code to reproduce all our experiments are available at `https://github.com/autogluon/tabrepo`

- A large scale evaluation of tabular models comprising 786000 model predictions with 1310 models from 10 different families densely evaluated across 200 datasets and 3 seeds.

- We show how the repository can be used to study at marginal cost the performance of tuning models while considering ensembles by leveraging precomputed model predictions.

- We show that TabRepo combined with transfer learning achieves state-of-the-art results compared to AutoML systems in accuracy and training time.

This paper first reviews related work before describing the TabRepo dataset. We then illustrate how TabRepo can be leveraged to compare HPO with ensemble against AutoML systems. Finally, we use transfer-learning to achieve a new state-of-the-art on tabular data.

## 2 Related work

Acquiring and re-using offline evaluations to eliminate redundant computation has been proposed in multiple compute intensive fields of machine learning. In HPO, several works proposed to acquire a large number of evaluations to simulate the performance of different optimizers across many seeds which can easily become prohibitive otherwise, in particular when the blackbox function optimized involves training a large neural network (Klein and Hutter, 2019; Eggensperger et al., 2021). Similarly, tabular benchmarks were acquired for Neural Architecture Search (Ying et al., 2019; Dong and Yang, 2020) as it was observed that, due to the large cost of comparisons, not enough seeds were used to distinguish methods properly from random-search (Yang et al., 2020).

While the cost of tabular methods can be orders of magnitude lower than training large neural networks, it can still be significant in particular when considering many methods, datasets, and seeds. Several works proposed to provide benchmarks with precomputed results, in particular Gorishniy et al. (2021); Grinsztajn et al. (2022); McElfresh et al. (2023). One key differentiator with those works is that TabRepo exposes model *predictions* which enables to simulate nearly instantaneously not only the errors of single models but also *ensembles* of any subset of available models. To the best of our knowledge, the only prior works that considered providing a dataset compatible with ensemble predictions is Borchert et al. (2022) in the context of time-series and Purucker and Beel (2022, 2023) in the context of tabular prediction. Our work differs from Purucker and Beel (2022, 2023) as they train a different number of models per dataset (sparse) and focus on evaluating the performance of ensembling algorithms, whereas we obtain results for all models on all datasets (dense) and use transfer-learning to create a model portfolio. Additionally they consider 31 classification datasets whereas we include 200 datasets both from regression and classification.

An important advantage of acquiring offline evaluations is that it allows to perform transfer-learning, e.g. to make use of the offline data to speed up the tuning of model hyperparameters. In particular, a popular transfer-learning approach is called Portfolio learning, or Zeroshot HPO, and consists of selecting greedily a set of models that are complementary and are then likely to perform well on a new dataset (Xu et al., 2010). Due to its performance and simplicity, the method has been applied in a wide range of applications ranging from HPO (Wistuba et al., 2015), time-series (Borchert et al., 2022), computer vision (Arango et al., 2023), tabular deep-learning (Winkelmolen et al., 2020; Zimmer et al., 2021), and AutoML (Feurer et al., 2015, 2022).

The current state-of-the-art for tabular methods in terms of accuracy is arguably AutoGluon (Erickson et al., 2020) in light of recent large scale benchmarks (Gijsbers et al., 2024). The method trains models from different families with bagging: each model is trained on several distinct non-overlapping random splits of the training dataset to generate out-of-fold predictions whose scores are likely to align well with performance on the test set. Then, another layer of models is trained whose inputs are both the original inputs concatenated with the predictions of the models in the previous layers. Finally, an ensemble is built on top of the last layer model predictions using ensemble selection (Caruana et al., 2004). Interestingly, this work showed that excellent

performance could be achieved without performing HPO and instead using a fixed list of manually selected model configurations. However, the obtained solution can be expensive for inference due to the use of model stacking and requires human experts to select default model configurations. Our work shows that using TabRepo, one can alleviate both caveats by learning default configurations which improves accuracy and latency when matching compute budget.

## 3  TabRepo

We now describe TabRepo and our notations to define its set of evaluations and predictions. In what follows, we denote $[n] = \{1, \ldots, n\}$ to be the set of the first $n$ integers.

**Model bagging**. All models are trained with *bagging* to better estimate their hold-out performance and improve their accuracy. Given a dataset split into a training set $(X^{(\text{train})}, y^{(\text{train})})$ and a test set $(X^{(\text{test})}, y^{(\text{test})})$ and a model $f^\lambda$ with parameters $\lambda$, we train $\mathcal{B}$ models on $\mathcal{B}$ non-overlapping cross-validation splits of the training set denoted $\{(X^{(\text{train})}[b], y^{(\text{train})}[b]), (X^{(\text{val})}[b], y^{(\text{val})}[b])\}_{b=1}^{\mathcal{B}}$. Each of the $\mathcal{B}$ model parameters are fitted by ERM, i.e. by minimizing the loss

$$\lambda_b = \arg\min_\lambda \mathcal{L}(f^\lambda(X^{(\text{train})}[b]), y^{(\text{train})}[b]), \quad \text{for } b \in [\mathcal{B}].$$

where the loss $\mathcal{L}$ is calculated via root mean-squared error (RMSE) for regression, the area under the receiver operating characteristic curve (AUC) for binary classification and log loss for multi-class classification. We choose these evaluation metrics to be consistent with AMLB defaults (Gijsbers et al., 2024).

One can then construct *out-of-fold predictions*[3] denoted as $\tilde{y}^{(\text{train})}$ that are computed on unseen data for each bagged model, i.e. predictions are obtained by applying the model on the validation set of each split i.e. $f^{\lambda_b}(X^{(\text{val})}[b])$ which allows to estimate the performance on the training set for unseen data. To predict on a test dataset $X^{(\text{test})}$, we average the predictions of the $\mathcal{B}$ fitted models,

$$\tilde{y}^{(\text{test})} = \frac{1}{\mathcal{B}} \sum_{b=1}^{\mathcal{B}} f^{\lambda_b}(X^{(\text{test})}). \tag{1}$$

**Datasets, predictions and evaluations**. We collect evaluations on $\mathcal{D} = 200$ datasets from OpenML (Vanschoren et al., 2014). For selecting the datasets, we combined two prior tabular dataset suites. The first is from the AutoMLBenchmark (Gijsbers et al., 2024), and the second is from the Auto-Sklearn 2 paper (Feurer et al., 2022). Refer to Appendix C for a detailed description of the datasets.

For each dataset, we generate $\mathcal{S} = 3$ tasks by selecting the first three of ten cross-validation fold as defined in OpenML's evaluation procedure, resulting in $\mathcal{T} = \mathcal{D} \times \mathcal{S}$ tasks in total. The list of $\mathcal{T}$ tasks' features and labels are denoted

$$\{((X_i^{(\text{train})}, y_i^{(\text{train})}), (X_i^{(\text{test})}, y_i^{(\text{test})}))\}_{i=1}^{\mathcal{T}}$$

where $X_i^s \in \mathbb{R}^{\mathcal{N}_i^s \times d_i}$ and $y_i \in \mathbb{R}^{\mathcal{N}_i^s \times o_i}$ for each split $s \in \{\text{train}, \text{test}\}$, $\mathcal{N}_i^s$ denotes the number of rows available in each split. Feature and label dimensions are denoted with $d_i$ and $o_i$ respectively. We use a loss $\mathcal{L}_i$ for each task depending on its type, in particular we use AUC for binary classification, log loss for multi-class classification and RMSE for regression.

For each task, we fit each model on $\mathcal{B} = 8$ cross-validation splits before generating predictions with Eq. 1. The predictions on the training and test splits for any task $i \in [\mathcal{T}]$ and model $j \in [\mathcal{M}]$ are denoted as

$$\tilde{y}_{ij}^{(\text{train})} \in \mathbb{R}^{\mathcal{N}_i^{(\text{train})} \times o_i}, \qquad \tilde{y}_{ij}^{(\text{test})} \in \mathbb{R}^{\mathcal{N}_i^{(\text{test})} \times o_i}. \tag{2}$$

---

[3]Note that for classification tasks, we refer to *prediction probabilities* as simply *predictions* for convenience.

We can then obtain losses for all tasks and models with

$$\ell_{ij}^{\text{(train)}} = \mathcal{L}_i(\tilde{y}_{ij}^{\text{(train)}}, y_i^{\text{(train)}}), \qquad \ell_{ij}^{\text{(test)}} = \mathcal{L}_i(\tilde{y}_{ij}^{\text{(test)}}, y_i^{\text{(test)}}). \tag{3}$$

For all tasks and models, we use the AutoGluon featurizer to preprocess the raw data prior to fitting the models (Erickson et al., 2020).

**Models available**. For base models, we consider Linear models, KNearestNeighbors, RandomForest (Breiman, 2001), ExtraTrees (Geurts et al., 2006), XGBoost (Chen and Guestrin, 2016), LightGBM (Ke et al., 2017), CatBoost (Prokhorenkova et al., 2018), and Multi-layer perceptron (MLP). We evaluate all *default* configurations used by AutoGluon for those base models together with 50 random configurations for Linear models and KNN and 200 random configurations for the remaining families. We also include a single default configuration each of TabPFN (Hollmann et al., 2022) and FT-Transformer (Gorishniy et al., 2021) which use GPUs to accelerate training. In total, this yields $\mathcal{M} = 1310$ configurations. All configurations are run for one hour. For the models that are not finished in one hour, we early stop them and use the best checkpoint according to the validation score to generate predictions.

In addition, we evaluate 6 AutoML frameworks: Auto-Sklearn 1 and 2 (Feurer et al., 2015, 2022), FLAML (Wang et al., 2021), LightAutoML (Vakhrushev et al., 2021), H2O AutoML (LeDell and Poirier, 2020) and AutoGluon (Erickson et al., 2020). We run all model configurations and AutoML frameworks via the AutoMLBenchmark (Gijsbers et al., 2024) for both 1h and 4h fitting time budget, using the implementations provided by the AutoML system authors.

For every task and model combination, we store losses defined in Eq. 3 and predictions defined in Eq. 2. Storing evaluations for every ensemble would be clearly infeasible given the large set of base models considered. However, given that we also store base model predictions, an ensemble can be fit and evaluated on the fly for any set of configurations by querying lookup tables as we will now describe.

**Ensembling**. Given the predictions from a set of models on a given task, we build ensembles by using the Caruana et al. (2004) approach. The procedure selects models by iteratively picking the model such that the average of selected models' predictions minimizes the error. Formally, given $\mathcal{M}$ model predictions $\{\tilde{y}_1, \ldots, \tilde{y}_{\mathcal{M}}\} \in \mathbb{R}^{\mathcal{M}}$, the strategy selects $\mathcal{C}$ models $j_1, \ldots, j_{\mathcal{C}}$ iteratively as follows

$$j_1 = \underset{j_1 \in \mathcal{M}}{\arg\min} \, \mathcal{L}(\tilde{y}_{j_1}, y^{\text{(train)}}), \qquad j_n = \underset{j_n \in \mathcal{M}}{\arg\min} \, \mathcal{L}(\frac{1}{n} \sum_{c=1}^{n} \tilde{y}_{j_c}, y^{\text{(train)}}).$$

The final predictions are obtained by averaging the selected models $j_1, \ldots, j_{\mathcal{C}}$:

$$\frac{1}{\mathcal{C}} \sum_{c=1}^{\mathcal{C}} \tilde{y}_{j_c}. \tag{4}$$

Note that the sum is performed over the vector of model indices which allow to potentially select a model multiple times and justifies the term "weight".

Critically, the performance of any ensemble of configurations can be calculated by summing the predictions of base models obtained from lookup tables. This is particularly fast as it does not require any retraining but only recomputing losses between weighted predictions and target labels. While we could leverage other ensembling methods in future, we prioritize Caruana et al. (2004) in this work due to its widespread adoption, strong performance, and fast training time.

## 4 Comparing HPO and AutoML systems

We now show how TabRepo can be used to analyze the performance of base model families and the effect of tuning hyperparameters with ensembling against recent AutoML systems. All experiments are done at marginal costs given that they just require querying precomputed evaluations and predictions.

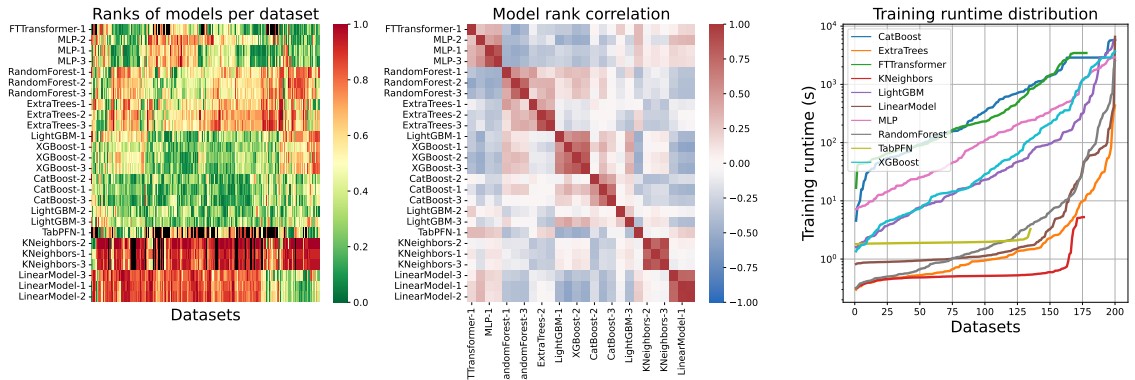

Figure 1: Cluster map of model rank for all datasets (left), correlation of model ranks (middle) and average runtime distribution over every dataset (right). For readability, only the first 3 configurations of each model family are displayed in the left and middle figures. The performance of models that could not be fitted successfully are represented in black.

**Model error and runtime distributions**. In Fig. 1, we start by analyzing the performance of different base models[4]. In particular, the rank of model losses over datasets shows that while some model families dominate in performance on aggregate such as gradient boosted methods CatBoost and LightGBM, in some tasks MLP are better suited. Looking at model correlations, we see interesting patterns as some model families are negatively correlated between each other such as MLP and XGBoost which hints at the potential benefit of ensembling.

Next, we plot the distribution of runtime configurations over all 200 datasets. We see that an order of magnitude separates respectively the training runtime of CatBoost from MLP, XGBoost and LightGBM, with the remaining methods being faster still. Importantly, while CatBoost obtains the strongest average rank among model families, it is also the most expensive which is an important aspect to take into account when considering possible training runtime constraints as we will see later in our experiments. We provide further analysis of model runtimes in Appendix H.

**Effect of tuning and ensembling on model error**. We now compare methods across all tasks by using both ranks and normalized errors. Ranks are computed over the $\mathcal{M}$ different models and all AutoML frameworks. Normalized errors are computed by reporting the relative distance to a topline loss compared to a baseline with $\frac{l_{\text{method}} - l_{\text{topline}}}{l_{\text{baseline}} - l_{\text{topline}}}$ while clipping the denominator to 1e-5 and the final score value to $[0, 1]$. We use respectively the top and median score among all scores to set the topline and baseline. The median allows to avoid having scores collapse when one model loss becomes very high which can happen frequently for regression cases in presence of overfitting or numerical instabilities.

**Comparison**. In Fig. 2, we show the aggregate of the normalized error across all tasks. For each model family, we evaluate the default hyperparameters, the best hyperparameter obtained after a random search of 4 hours and an ensemble built on top of the best 20 configurations obtained by this search. In Fig. 2, we see that CatBoost dominates other models while FT-Transformer and LightGBM are the runner-ups. Tuning model hyperparameters and ensembling improves all models and ensembling allows LightGBM to match CatBoost's accuracy. No model is able to beat state-of-the-art AutoML systems even with tuning and ensembling. This is unsurprising as all state-of-the-art tabular methods considered multiple model families in order to reach good performance and echoes the finding of Erickson et al. (2020).

---

[4]All models considered run successfully except for TabPFN, FT-Transformer and KNN where failures are depicted in black in Fig. 1 left which are due to memory or implementation issues.

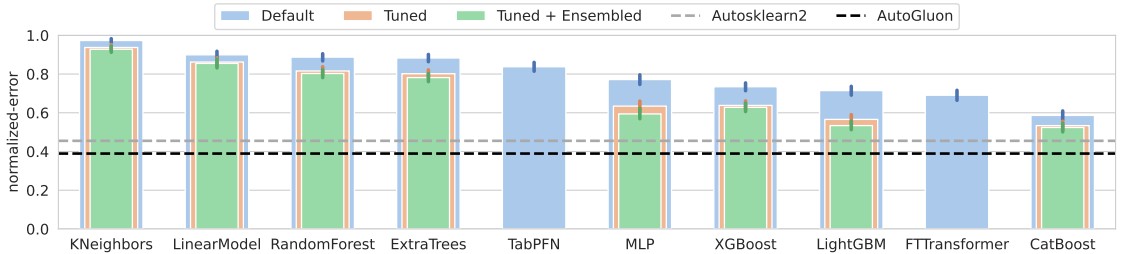

Figure 2: Normalized error for all model families when using default hyperparameters, tuned hyperparameters, and ensembling after tuning. All methods are run with a 4h budget.

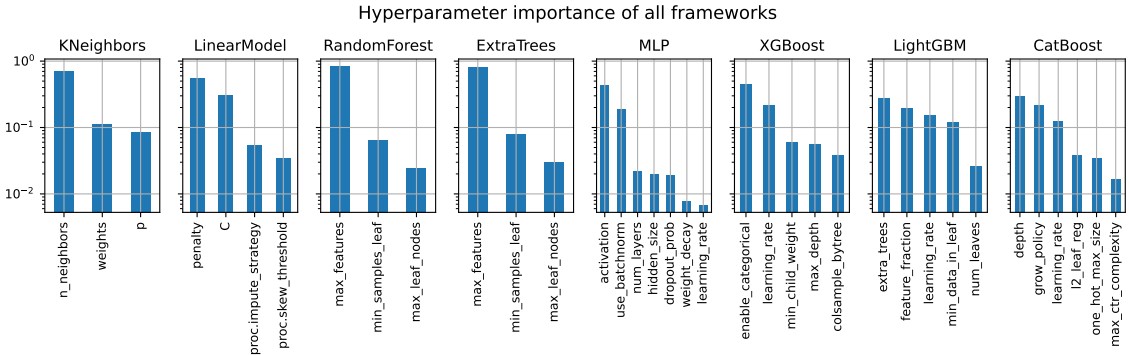

Figure 3: Hyperparameter importance for each model family using the fANOVA method from Hutter et al. (2014). Y-axis is in log-space.

**Hyperparameters importance**. We just saw how tuning and ensembling can improve the performance of each model family. In Fig. 3, we show the importance of hyperparameters for each family that can be evaluated on CPU. We use the fANOVA method proposed by Hutter et al. (2014) on each task and numerize features to fit random forest models using the search space provided in section E.1. The importance are computed on each task and then aggregated.

We observe that hyperparameter importance mostly follows practitioner intuition. For instance, the number of neighbors, the regularization, and the depth are the top hyperparameters respectively for KNN, Linear model and CatBoost as expected. For MLP, the most important hyperparameters are the activation and whether to use batch-normalization. For this model family, some options such as using batch-normalization dominate. One could consider model-based or multi-fidelity approaches to generate configurations for model families as opposed to random-search as proposed in (Winkelmolen et al., 2020) to exploit parts of the search-space that are better suited.

## 5 Portfolio learning with TabRepo

We just saw how TabRepo can be leveraged to analyze the performance of frameworks when performing tuning and ensembling. In particular, we saw that ensembling a model family after tuning does not outperform current AutoML systems. We now show how TabRepo can be combined with transfer learning techniques to perform the tuning search offline and outperform current AutoML methods.

**Portfolio learning**. To leverage offline data and speed-up model selection, Xu et al. (2010) proposed an approach to learn a portfolio of complementary configurations that performs well on average when evaluating all the configurations of the portfolio and selecting the best one.

Table 1: Normalized error, rank, training and inference time averaged over all tasks given 4h training budget. Inference time is calculated as the prediction time on the test data divided by the number of rows in the test data.

| method | normalized-error | rank | time fit (s) | time infer (s) |
|---|---|---|---|---|
| **Portfolio (ensemble) (ours)** | **0.365** | **168.7** | 6275.5 | 0.050 |
| AutoGluon | 0.389 | 208.2 | 5583.1 | 0.062 |
| Portfolio (ours) | 0.434 | 232.5 | 6275.5 | 0.012 |
| Autosklearn2 | 0.455 | 243.5 | 14415.9 | 0.013 |
| Lightautoml | 0.466 | 246.1 | 9173.9 | 0.298 |
| Flaml | 0.513 | 317.8 | 14267.0 | 0.002 |
| CatBoost (tuned + ensemble) | 0.524 | 267.3 | 9065.2 | 0.012 |
| H2oautoml | 0.526 | 337.0 | 13920.1 | 0.002 |
| CatBoost (tuned) | 0.534 | 284.7 | 9065.2 | 0.002 |
| LightGBM (tuned + ensemble) | 0.534 | 268.7 | 3528.9 | 0.010 |
| LightGBM (tuned) | 0.566 | 304.2 | 3528.9 | 0.001 |
| CatBoost (default) | 0.586 | 341.2 | 456.8 | 0.002 |
| MLP (tuned + ensemble) | 0.594 | 402.5 | 5771.8 | 0.098 |
| XGBoost (tuned + ensemble) | 0.628 | 357.9 | 4972.7 | 0.013 |
| MLP (tuned) | 0.634 | 451.9 | 5771.8 | 0.014 |
| XGBoost (tuned) | 0.638 | 376.5 | 4972.7 | 0.002 |
| FTTransformer (default) | 0.690 | 532.1 | 567.4 | 0.003 |
| LightGBM (default) | 0.714 | 491.5 | 55.7 | 0.001 |
| XGBoost (default) | 0.734 | 522.2 | 75.1 | 0.002 |
| MLP (default) | 0.772 | 629.4 | 38.2 | 0.015 |
| ExtraTrees (tuned + ensemble) | 0.782 | 544.2 | 538.3 | 0.001 |
| ExtraTrees (tuned) | 0.802 | 572.5 | 538.3 | 0.000 |
| RandomForest (tuned + ensemble) | 0.803 | 578.3 | 1512.2 | 0.001 |
| RandomForest (tuned) | 0.816 | 598.0 | 1512.2 | 0.000 |
| TabPFN (default) | 0.837 | 731.9 | 3.8 | 0.016 |
| LinearModel (tuned + ensemble) | 0.855 | 873.8 | 612.4 | 0.038 |
| LinearModel (tuned) | 0.862 | 891.6 | 612.4 | 0.006 |
| ExtraTrees (default) | 0.883 | 788.6 | 3.0 | 0.000 |
| RandomForest (default) | 0.887 | 773.9 | 13.8 | 0.000 |
| LinearModel (default) | 0.899 | 940.1 | 7.1 | 0.014 |
| KNeighbors (tuned + ensemble) | 0.928 | 980.8 | 12.0 | 0.001 |
| KNeighbors (tuned) | 0.937 | 1016.5 | 12.0 | 0.000 |
| KNeighbors (default) | 0.973 | 1149.1 | 0.6 | 0.000 |

Similarly to Caruana ensemble selection described in Eq. 4, the method iteratively selects $\mathcal{N} < \mathcal{M}$ configurations as follows

$$j_1 = \underset{j_1 \in [\mathcal{M}]}{\arg\min} \, \mathbb{E}_{i \sim [\mathcal{T}]}[\ell_{ij_1}^{(\text{train})}], \qquad j_n = \underset{j_n \in [\mathcal{M}]}{\arg\min} \, \mathbb{E}_{i \sim [\mathcal{T}]}[\underset{k \in [n]}{\min} \, \ell_{ij_k}^{(\text{train})}].$$

At each iteration, the method greedily picks the configuration that has the lowest average error when combined with previously selected configurations.

**Anytime portfolio**. Fitting portfolio configurations can be done in an *any-time* fashion given a fitting time budget. To do so, we evaluate portfolio configurations sequentially until the budget is exhausted and use only models trained up to this point to select an ensemble. In cases where the first configuration selected by the portfolio takes longer to run than the constraint, we instead report the result of a fast baseline as in Gijsbers et al. (2024).

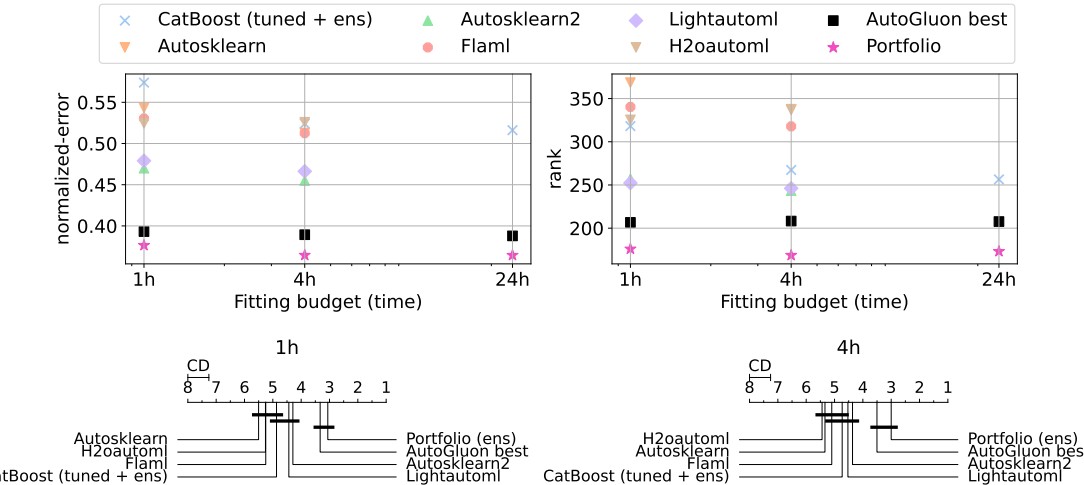

Figure 4: Top: scatter plot of average normalized error (left) and rank (right) against fitting training time budget. Bottom: Critical difference (CD) diagram showing average rank between method selected and which methods are tied statistically by a horizontal bar.

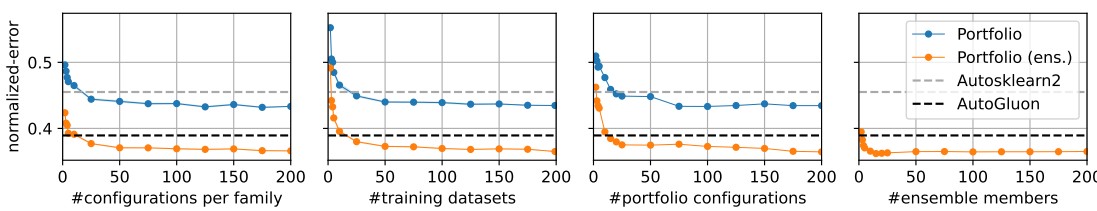

Figure 5: Impact on normalized error when varying the (a) number of configurations per family, (b) number of training datasets, (c) portfolio size and (d) number of ensemble members.

**Evaluations**. We evaluate the anytime portfolio approach in a standard leave-one-out setting. When evaluating on the $i$-th dataset, we compute portfolio configurations on $\mathcal{D} - 1$ training datasets by excluding the $i$-th test dataset to avoid potential leakage.

Results are reported in Tab. 1 when considering a 4h fitting budget constraint. Given that all AutoML baselines require only CPUs, we consider only CPU models and report results with portfolios including TabPFN and FT-Transformer separately in Appendix I.

We report both the performance of the best model according to validation error (Portfolio) and when ensembling the selected portfolio configurations (Portfolio (ensemble)). The portfolio combined with ensembling outperforms AutoGluon for both accuracy and latency given the same 4h fitting budget even without stacking. When picking the best model without ensembling, the portfolio still retains good performance and outperforms all frameworks other than AutoGluon.

In Fig. 4, we report the performance for different fitting budgets. Ensembles of portfolio configurations can beat all AutoML frameworks for all metrics for 1h, 4h and 24h budget without requiring stacking which allows to obtain a lower latency compared to AutoGluon. Critical difference (CD) diagrams from Demšar (2006) show that while Portfolio has better aggregate performance than other methods, AutoGluon and Portfolio are tied statistically and are the only methods that are statistically better than all baselines. Interestingly, a baseline consisting of tuning and ensembling CatBoost models is surprisingly strong and outperforms multiple AutoML methods. As in the previous section, all evaluations are obtained from pre-computed results in TabRepo.

**How much data is needed?** We have seen that TabRepo allows to learn portfolio configurations that can outperform state-of-the-art AutoML systems. Next, we analyze the question of how much data is needed for transfer learning to achieve strong results in two dimensions, namely: how many offline configurations and datasets are required to reach good performance? While important, these dimensions are rarely analyzed in previous transfer learning studies due to their significant cost.

In Fig. 5, we vary both of those dimensions independently with a budget of 4h for all methods. When evaluating on a test dataset, we pick a random subset of configurations $\mathcal{M}'$ per model family in the first case (5a) and a random subset of $\mathcal{D}' < \mathcal{D}$ datasets in the second case (5b) and report mean and standard error over 10 different seeds. Portfolio with ensembling starts outperforming AutoGluon at around 10 configurations or training datasets. Having more datasets or more configurations in offline data both improve the final performance up to a certain point with a saturating effect around 150 offline configurations or offline datasets.

Next, we investigate the impact of the portfolio size in Fig. 5c. We observe that a portfolio of size 3 is sufficient to outperform all AutoML systems except AutoGluon, and a portfolio of size 15 is sufficient to outperform AutoGluon, with consistent benefit with increasing portfolio size. An in-depth analysis of model family pick order in the Portfolio is provided in Appendix G.

Finally, we investigate the impact of the number of selected models in the ensemble defined in Eq. 4 in Fig. 5d. While AutoML systems typically use large values (for instance AutoGluon uses 100 iterations), we demonstrate that the best performance is achieved at a mere 15 iterations, with more iterations having no measurable improvement.

## 6 Conclusion

In this paper, we introduced TabRepo, a benchmark of tabular models on a large number of datasets. Critically, the repository contains not only model evaluations but also predictions which allows to efficiently evaluate ensemble strategies. We showed that the benchmark can be used to analyze the performance of different tuning strategies combined with ensembling at marginal cost. We also showed how the dataset can be used to learn portfolio configurations that outperforms state-of-the-art tabular methods for accuracy, training time and latency.

The repository can also be used to improve real-world systems. To illustrate this, we report the 2023 AMLB results (Gijsbers et al., 2024) in Tab. 2. In addition, we evaluated AutoGluon using portfolio configurations learned from TabRepo. Combining AutoGluon with a fixed learned portfolio outper-

Table 2: Performance of AutoGluon combined with portfolios on AMLB.

| method | win-rate | loss reduc. |
|---|---|---|
| **AG + Portfolio (ours)** | - | **0%** |
| AG | 67% | 2.8% |
| MLJAR | 81% | 22.5% |
| lightautoml | 83% | 11.7% |
| GAMA | 86% | 15.5% |
| FLAML | 87% | 16.3% |
| autosklearn | 89% | 11.8% |
| H2OAutoML | 92% | 10.3% |
| CatBoost | 94% | 18.1% |
| TunedRandomForest | 94% | 22.9% |
| RandomForest | 97% | 25.0% |
| XGBoost | 98% | 20.9% |
| LightGBM | 98% | 23.6% |

forms all current systems with a win-rate of 67% compared to the best system, establishing a new state-of-the-art. This illustrates a key application of TabRepo: to improve AutoML systems. AutoGluon 1.0 has adopted TabRepo's learned portfolio as its new default[5], replacing the prior hand-crafted default configuration.

The code for accessing evaluations from TabRepo and evaluating any ensemble together with the scripts used to generate all the paper results are available in the GitHub repository. We hope this paper will facilitate future research on new methods combining ideas from CASH, multi-fidelity and transfer-learning to further improve the state-of-the-art in tabular prediction.

---

[5]AutoGluon 1.0 Portfolio | AutoGluon 1.0 TabRepo Release Highlight

## 7 Broader Impact Statement

Generating initial results for offline configurations is expensive. In total, 219136 CPU hours were needed to complete all model evaluations of TabRepo. However, performing the analysis done in this paper without leveraging precomputed evaluations and predictions would have required 4066576 CPU hours which is $\sim 18.6$ times more expensive. Using precomputed predictions from TabRepo enables researchers to obtain results requiring model interactions (such as ensembling) 10000x cheaper than retraining the base models from scratch. We hope that the research community will use TabRepo to experiment on more ideas which would further amortize its cost.

Collecting results on more models and datasets improves final performance as seen in Fig. 5. This incentivizes researchers to use even larger collections of datasets which makes checking the ethical aspect of each dataset more challenging. This is a burning issue in the case of foundation models which requires gathering large datasets and is in tension with checking thoroughly ethical aspects for all datasets (Birhane et al., 2021). For TabRepo, we checked the license and ethic aspects for each dataset but finding a solution to verify a large collection of datasets is an open problem.

One limitation of our work is that we are not taking advantage of dataset features as opposed to previous work (Jomaa et al., 2021). We believe future work could leverage our repository and improve results further by looking at advanced techniques to exploit dataset statistics in the context of ensembling.

Finally, we considered random configurations for each model family which restricts the performance given that model-based approaches can be more efficient. Future work could investigate more efficient approaches such as model-based or multi-fidelity as in Winkelmolen et al. (2020) to generate more performant configurations for all model families.

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

## A  Additional experiment details

**Number of Caruana Steps**. In all our experiments, we set the number of Caruana steps to $\mathcal{C} = 40$ when building ensembles of base models or portfolio configurations. We observe that values beyond 40 provide negligible benefit while linearly increasing the runtime of simulations in TabRepo. We also note that in our ablations in Fig. 5d we observe that $\mathcal{C} = 15$ achieves the best normalized-error value, and values between 25 and 200 achieve nearly identical results.

**Fallback method**. We use the default configuration of Extra-trees as a backup when the first configuration of a portfolio does not finish under the constraint which takes just a few seconds to evaluate.

**Number of portfolio configurations**. When reporting results on a portfolio, we apply the anytime procedure described in Sec 5 and run at most $\mathcal{N} = 200$ portfolio configurations. Setting this bound serves mostly as an upper-bound as all configurations are almost never evaluated given that all configurations have to be trained under the fitting budget.

**Hardware details**. All model configuration and AutoML framework results were obtained on AWS EC2 machines via AutoMLBenchmark's AWS mode functionality. For all model and AutoML evaluations, we used m6i.2xlarge EC2 instances with 100 GB of gp3 storage. These instances have 8 virtual CPUs (4 physical CPUs) and 32 GB of memory. The Python version used for all experiments was 3.9.18. We chose m6i.2xlarge instances to align with AutoMLBenchmark's choice of m5.2xlarge instances. m5 instances have the same number of CPUs and memory, but the m6i instances were more cost-efficient due to faster CPUs.

All simulation paper experiments in Sec. 4 and 5 were done on a m6i.32xlarge and takes less than 3 hours of compute. It can also be done with an m6i.4xlarge which takes less than 24 hours.

**Critical difference diagrams**. We use Autorank (Herbold, 2020) to compute critical difference diagrams.

**Data-structure**. TabRepo takes 107 GB on disk. To avoid requiring large memory cost, we use a memmap data-structure which loads model evaluations on the fly from disk to memory when needed. This allows to reduce the RAM requirement to ~20GB of RAM.

## B  API

In Listing 1, we show an example of calling TabRepo to retrieve the ensemble performance of a list of models. Because we store all model predictions, we are able to reconstruct the metrics of any ensemble among the $\mathcal{M} = 1310$ models considered.

```python
from tabrepo import EvaluationRepository

# load TabRepo with 200 datasets, 3 folds and 1530 configurations
repository = EvaluationRepository.from_context(version="D244_F3_C1530_200")

# returns in ~2s the tensor of metrics for each dataset/fold obtained after
    ensembling the given configurations
metrics = repository.evaluate_ensemble(
    datasets=["balance-scale", "page-blocks"],  # dataset to report results on
    folds=[0, 1, 2],  # which folds to consider for each dataset
    configs=["CatBoost_r42_BAG_L1", "NeuralNetTorch_r40_BAG_L1"],  # configs that
    are ensembled
    ensemble_size=40,  # maximum number of Caruana steps
)

# returns the predictions on the val data for a given task and config
```

```
15  val_predictions = repository.predict_val(
16      dataset="page-blocks", fold=2, config="ExtraTrees_r7_BAG_L1"
17  )
18
19  # returns the predictions on the test data for a given task and config
20  test_predictions = repository.predict_test(
21      dataset="page-blocks", fold=2, config="ExtraTrees_r7_BAG_L1"
22  )
```

Listing 1: Example of calling TabRepo to obtain performance scores on an ensemble configuration or model predictions on validation/test splits.

## C Dataset details

For selecting the datasets, we combined two prior tabular dataset suites. The first is from AutoML-Benchmark (Gijsbers et al., 2024), containing 104 datasets. The second is from the Auto-Sklearn 2 paper (Feurer et al., 2022), containing 208 datasets.

All datasets are publicly available via OpenML. After de-duplicating, the union contains 289 datasets. The AutoMLBenchmark datasets have been previously filtered from a larger set via a specific inclusion criteria detailed in section 5.1.1 of Gijsbers et al. (2024). Notably, they filter out datasets that are trivial, such that simple methods such as a random forest cannot perfectly solve them. We perform a similar operation by fitting a default Random Forest configuration on all 289 datasets and filtering any dataset that is trivial (AUC > 0.999, log loss < 0.001, or r2 > 0.999). After filtering trivial datasets, we are left with 244 datasets.

We then run all AutoML baselines and model configurations on the 244 datasets (3 folds, for a total of 732 tasks). We performed re-runs on failed tasks when necessary to attempt to get results on all models and AutoML systems for every dataset, but sometimes this was not possible due to problems such as out-of-memory errors or AutoML system implementation errors outside our control. For datasets with model or AutoML system failures, we exclude them. We exclude datasets rather than impute missing values to ensure the results being computed are fully verifiable and replicable in practice. After excluding datasets with model or AutoML system failures, we have 211 datasets remaining.

We make an exception to this exclusion criteria for the following cases:

- KNN fails due to the dataset not having numeric features.

- TabPFN fails due to the dataset having greater than 100 features or greater than 10 classes.

- FT-Transformer fails due to out-of-memory errors.

In the above special cases, we instead keep the dataset and will impute the result of the failed model using our default Extra-trees configuration.

Finally, we filter out the 11 largest datasets for practical usability purposes of TabRepo. This is because loading the prediction probabilities of 1310 model configurations on large (multi-class) datasets leads to significant challenges. As an example, the total size of the predictions for 211 datasets is 455 GB. By reducing to 200 datasets, the size decreases dramatically to 107 GB (The full 244 datasets is 4.5 TB).

In total, we use 105 binary classification datasets, 68 multi-class classification datasets and 27 regression datasets. We provide a table of dataset summary statistics in Tab. 3 and an exhaustive list of the 200 datasets used in TabRepo separated by problem type in Tab. 4, Tab. 5 and Tab. 6 where we list for each dataset the TaskID OpenML identifier, the dataset name, the number of rows $n$, the number of features $f$ and the number of classes $C$ which is always 2 for binary classification.

Table 3: Statistics of the 200 datasets in TabRepo

|      | n      | f     |
|------|--------|-------|
| mean | 15097  | 570   |
| std  | 38065  | 2161  |
| min  | 100    | 3     |
| 5%   | 500    | 3     |
| 10%  | 575    | 5     |
| 25%  | 1107   | 10    |
| 50%  | 3800   | 20    |
| 75%  | 10027  | 60    |
| 90%  | 41173  | 506   |
| 95%  | 70472  | 1787  |
| max  | 400000 | 10936 |

### C.1 Train-test splits

For all datasets we use the OpenML 10-fold Cross-validation estimation procedure and select the first 3 folds for our experiments. For each task (a particular dataset fold), we use 90% of the data as training and 10% as test. We use identical splits to Gijsbers et al. (2024).

## D  Raw Results and Reproducibility

- To reproduce all paper results and to get the predictions of all model configs on all tasks, follow the README instructions here: `https://github.com/autogluon/tabrepo`

- The raw results for all model configs on all tasks are available here: `https://tabrepo.s3.us-west-2.amazonaws.com/contexts/2023_11_14/configs.csv`

- The raw results for all baselines and AutoML systems on all tasks are available here: `https://tabrepo.s3.us-west-2.amazonaws.com/contexts/2023_11_14/baselines.csv`

- The raw results for all TabRepo experiments are available here: `https://tabrepo.s3.us-west-2.amazonaws.com/paper/automl2024/results.zip`

- To generate the raw results of new model configs on new datasets, refer to this example script: `https://github.com/autogluon/tabrepo/blob/main/examples/run_quickstart_from_scratch.py`

- To regenerate all of the pre-computed results for all model configs and AutoML baselines (minimum 500,000 CPU hours across 92,000 EC2 instances), refer to these instructions: `https://github.com/autogluon/tabrepo/blob/main/examples/run_quickstart_from_scratch.py`

- The script used to generate Fig. 2 is available here: `https://github.com/Innixma/autogluon-benchmark/blob/master/v1_results/run_eval_tabrepo_v1.py`

## E  Model details

For each model type, we used the latest available package versions when possible. The precise versions used for each model are documented in Tab. 7.

For each model family, we choose 201 configurations, 1 being the default hyperparameters, as well as 200 randomly selected hyperparameter configs.

Table 4: Binary classification datasets used in TabRepo.

| Task ID | name | n | f | C | Task ID | name | n | f | C |
|---|---|---|---|---|---|---|---|---|---|
| 3593 | 2dplanes | 40768 | 10 | 2 | 3783 | fri_c2_500_50 | 500 | 50 | 2 |
| 168868 | APSFailure | 76000 | 170 | 2 | 3606 | fri_c3_1000_10 | 1000 | 10 | 2 |
| 359979 | Amazon_employee_acce | 32769 | 9 | 2 | 3581 | fri_c3_1000_25 | 1000 | 25 | 2 |
| 146818 | Australian | 690 | 14 | 2 | 3799 | fri_c3_500_10 | 500 | 10 | 2 |
| 359967 | Bioresponse | 3751 | 1776 | 2 | 3800 | fri_c3_500_50 | 500 | 50 | 2 |
| 359992 | Click_prediction_sma | 39948 | 11 | 2 | 3608 | fri_c4_500_100 | 500 | 100 | 2 |
| 361331 | GAMETES_Epistasis_2- | 1600 | 1000 | 2 | 3764 | fried | 40768 | 10 | 2 |
| 361332 | GAMETES_Epistasis_2- | 1600 | 20 | 2 | 189922 | gina | 3153 | 970 | 2 |
| 361333 | GAMETES_Epistasis_2- | 1600 | 20 | 2 | 9970 | hill-valley | 1212 | 100 | 2 |
| 361334 | GAMETES_Epistasis_3- | 1600 | 20 | 2 | 3892 | hiva_agnostic | 4229 | 1617 | 2 |
| 361335 | GAMETES_Heterogeneit | 1600 | 20 | 2 | 3688 | houses | 20640 | 8 | 2 |
| 361336 | GAMETES_Heterogeneit | 1600 | 20 | 2 | 9971 | ilpd | 583 | 10 | 2 |
| 359966 | Internet-Advertiseme | 3279 | 1558 | 2 | 168911 | jasmine | 2984 | 144 | 2 |
| 359990 | MiniBooNE | 130064 | 50 | 2 | 3904 | jm1 | 10885 | 21 | 2 |
| 3995 | OVA_Colon | 1545 | 10935 | 2 | 359962 | kc1 | 2109 | 21 | 2 |
| 3976 | OVA_Endometrium | 1545 | 10935 | 2 | 3913 | kc2 | 522 | 21 | 2 |
| 3968 | OVA_Kidney | 1545 | 10935 | 2 | 3704 | kdd_el_nino-small | 782 | 8 | 2 |
| 3964 | OVA_Lung | 1545 | 10935 | 2 | 3844 | kdd_internet_usage | 10108 | 68 | 2 |
| 4000 | OVA_Ovary | 1545 | 10935 | 2 | 359991 | kick | 72983 | 32 | 2 |
| 3980 | OVA_Prostate | 1545 | 10936 | 2 | 3672 | kin8nm | 8192 | 8 | 2 |
| 359971 | PhishingWebsites | 11055 | 30 | 2 | 190392 | madeline | 3140 | 259 | 2 |
| 361342 | Run_or_walk_informat | 88588 | 6 | 2 | 9976 | madelon | 2600 | 500 | 2 |
| 359975 | Satellite | 5100 | 36 | 2 | 3483 | mammography | 11183 | 6 | 2 |
| 125968 | SpeedDating | 8378 | 120 | 2 | 3907 | mc1 | 9466 | 38 | 2 |
| 361339 | Titanic | 2201 | 3 | 2 | 3623 | meta | 528 | 21 | 2 |
| 190411 | ada | 4147 | 48 | 2 | 3899 | mozilla4 | 15545 | 5 | 2 |
| 359983 | adult | 48842 | 14 | 2 | 3749 | no2 | 500 | 7 | 2 |
| 3600 | ailerons | 13750 | 40 | 2 | 359980 | nomao | 34465 | 118 | 2 |
| 190412 | arcene | 100 | 10000 | 2 | 167120 | numerai28.6 | 96320 | 21 | 2 |
| 3812 | arsenic-female-bladd | 559 | 4 | 2 | 190137 | ozone-level-8hr | 2534 | 72 | 2 |
| 9909 | autoUniv-au1-1000 | 1000 | 20 | 2 | 361341 | parity5_plus_5 | 1124 | 10 | 2 |
| 359982 | bank-marketing | 45211 | 16 | 2 | 3667 | pbcseq | 1945 | 18 | 2 |
| 3698 | bank32nh | 8192 | 32 | 2 | 3918 | pc1 | 1109 | 21 | 2 |
| 3591 | bank8FM | 8192 | 8 | 2 | 3919 | pc2 | 5589 | 36 | 2 |
| 359955 | blood-transfusion-se | 748 | 4 | 2 | 3903 | pc3 | 1563 | 37 | 2 |
| 3690 | boston_corrected | 506 | 20 | 2 | 359958 | pc4 | 1458 | 37 | 2 |
| 359968 | churn | 5000 | 20 | 2 | 190410 | philippine | 5832 | 308 | 2 |
| 146819 | climate-model-simula | 540 | 20 | 2 | 168350 | phoneme | 5404 | 5 | 2 |
| 3793 | colleges_usnews | 1302 | 33 | 2 | 3616 | pm10 | 500 | 7 | 2 |
| 3627 | cpu_act | 8192 | 21 | 2 | 3735 | pollen | 3848 | 5 | 2 |
| 3601 | cpu_small | 8192 | 12 | 2 | 3618 | puma32H | 8192 | 32 | 2 |
| 168757 | credit-g | 1000 | 20 | 2 | 3681 | puma8NH | 8192 | 8 | 2 |
| 14954 | cylinder-bands | 540 | 39 | 2 | 359956 | qsar-biodeg | 1055 | 41 | 2 |
| 3668 | delta_ailerons | 7129 | 5 | 2 | 9959 | ringnorm | 7400 | 20 | 2 |
| 3684 | delta_elevators | 9517 | 6 | 2 | 3583 | rmftsa_ladata | 508 | 10 | 2 |
| 37 | diabetes | 768 | 8 | 2 | 43 | spambase | 4601 | 57 | 2 |
| 125920 | dresses-sales | 500 | 12 | 2 | 359972 | sylvine | 5124 | 20 | 2 |
| 9983 | eeg-eye-state | 14980 | 14 | 2 | 361340 | tokyo1 | 959 | 44 | 2 |
| 219 | electricity | 45312 | 8 | 2 | 9943 | twonorm | 7400 | 20 | 2 |
| 3664 | fri_c0_1000_5 | 1000 | 5 | 2 | 3786 | visualizing_soil | 8641 | 4 | 2 |
| 3747 | fri_c0_500_5 | 500 | 5 | 2 | 146820 | wilt | 4839 | 5 | 2 |
| 3702 | fri_c1_1000_50 | 1000 | 50 | 2 | 3712 | wind | 6574 | 14 | 2 |
| 3766 | fri_c2_1000_25 | 1000 | 25 | 2 | | | | | |

The search spaces used are based on the search spaces defined in AutoGluon. We expanded the search range of various hyperparameters for increased model variety. Note that selecting the appropriate search space is a complex problem, and is not the focus of this work. TabRepo is built to work with arbitrary model configurations, and we welcome the research community to improve upon our initial baselines.

For all models we re-use the AutoGluon implementation for data pre-processing, initial hyperparameters, training, and prediction. We do this because choosing the appropriate pre-processing logic for an individual model is complex and introduces a myriad of design questions and potential pitfalls.

For maximum training epochs / iterations, instead of searching for an optimal value directly, we instead rely on the early stopping logic implemented in AutoGluon which sets the iterations to 10,000 for gradient boosting models and epochs to 500 for MLP.

Table 5: Multi-class classification datasets used in TabRepo.

| Task ID | name | n | f | C | Task ID | name | n | f | C |
|---|---|---|---|---|---|---|---|---|---|
| 211986 | Diabetes130US | 101766 | 49 | 3 | 6 | letter | 20000 | 16 | 26 |
| 359970 | GesturePhaseSegmenta | 9873 | 32 | 5 | 359961 | mfeat-factors | 2000 | 216 | 10 |
| 360859 | Indian_pines | 9144 | 220 | 8 | 359953 | micro-mass | 571 | 1300 | 20 |
| 125921 | LED-display-domain-7 | 500 | 7 | 10 | 189773 | microaggregation2 | 20000 | 20 | 5 |
| 146800 | MiceProtein | 1080 | 81 | 8 | 359993 | okcupid-stem | 50789 | 19 | 3 |
| 361330 | Traffic_violations | 70340 | 20 | 3 | 28 | optdigits | 5620 | 64 | 10 |
| 168300 | UMIST_Faces_Cropped | 575 | 10304 | 20 | 30 | page-blocks | 5473 | 10 | 5 |
| 3549 | analcatdata_authorsh | 841 | 70 | 4 | 32 | pendigits | 10992 | 16 | 10 |
| 3560 | analcatdata_dmft | 797 | 4 | 6 | 359986 | robert | 10000 | 7200 | 10 |
| 14963 | artificial-character | 10218 | 7 | 10 | 2074 | satimage | 6430 | 36 | 6 |
| 9904 | autoUniv-au6-750 | 750 | 40 | 8 | 359963 | segment | 2310 | 19 | 7 |
| 9906 | autoUniv-au7-1100 | 1100 | 12 | 5 | 9964 | semeion | 1593 | 256 | 10 |
| 9905 | autoUniv-au7-700 | 700 | 12 | 3 | 359987 | shuttle | 58000 | 9 | 7 |
| 11 | balance-scale | 625 | 4 | 3 | 41 | soybean | 683 | 35 | 19 |
| 2077 | baseball | 1340 | 16 | 3 | 45 | splice | 3190 | 60 | 3 |
| 359960 | car | 1728 | 6 | 4 | 168784 | steel-plates-fault | 1941 | 27 | 7 |
| 9979 | cardiotocography | 2126 | 35 | 10 | 3512 | synthetic_control | 600 | 60 | 6 |
| 359959 | cmc | 1473 | 9 | 3 | 125922 | texture | 5500 | 40 | 11 |
| 359957 | cnae-9 | 1080 | 856 | 9 | 190146 | vehicle | 846 | 18 | 4 |
| 146802 | collins | 1000 | 23 | 30 | 9924 | volcanoes-a2 | 1623 | 3 | 5 |
| 359977 | connect-4 | 67557 | 42 | 3 | 9925 | volcanoes-a3 | 1521 | 3 | 5 |
| 168909 | dilbert | 10000 | 2000 | 5 | 9926 | volcanoes-a4 | 1515 | 3 | 5 |
| 359964 | dna | 3186 | 180 | 3 | 9927 | volcanoes-b1 | 10176 | 3 | 5 |
| 359954 | eucalyptus | 736 | 19 | 5 | 9928 | volcanoes-b2 | 10668 | 3 | 5 |
| 3897 | eye_movements | 10936 | 27 | 3 | 9931 | volcanoes-b5 | 9989 | 3 | 5 |
| 168910 | fabert | 8237 | 800 | 7 | 9932 | volcanoes-b6 | 10130 | 3 | 5 |
| 359969 | first-order-theorem- | 6118 | 51 | 6 | 9920 | volcanoes-d1 | 8753 | 3 | 5 |
| 14970 | har | 10299 | 561 | 6 | 9923 | volcanoes-d4 | 8654 | 3 | 5 |
| 3481 | isolet | 7797 | 617 | 26 | 9915 | volcanoes-e1 | 1183 | 3 | 5 |
| 211979 | jannis | 83733 | 54 | 4 | 359985 | volkert | 58310 | 180 | 10 |
| 359981 | jungle_chess_2pcs_ra | 44819 | 6 | 3 | 9960 | wall-robot-navigatio | 5456 | 24 | 4 |
| 9972 | kr-vs-k | 28056 | 6 | 18 | 58 | waveform-5000 | 5000 | 40 | 3 |
| 2076 | kropt | 28056 | 6 | 18 | 361345 | wine-quality-red | 1599 | 11 | 6 |
| 361344 | led24 | 3200 | 24 | 10 | 359974 | wine-quality-white | 4898 | 11 | 7 |

Table 6: Regression datasets used in TabRepo.

| Task ID | name | n | f | Task ID | name | n | f |
|---|---|---|---|---|---|---|---|
| 233212 | Allstate_Claims_Seve | 188318 | 130 | 359952 | house_16H | 22784 | 16 |
| 359938 | Brazilian_houses | 10692 | 12 | 359951 | house_prices_nominal | 1460 | 79 |
| 360945 | MIP-2016-regression | 1090 | 144 | 359949 | house_sales | 21613 | 21 |
| 233215 | Mercedes_Benz_Greene | 4209 | 376 | 359946 | pol | 15000 | 48 |
| 167210 | Moneyball | 1232 | 14 | 359930 | quake | 2178 | 3 |
| 359941 | OnlineNewsPopularity | 39644 | 59 | 359931 | sensory | 576 | 11 |
| 359948 | SAT11-HAND-runtime-r | 4440 | 116 | 359932 | socmob | 1156 | 5 |
| 317614 | Yolanda | 400000 | 100 | 359933 | space_ga | 3107 | 6 |
| 359944 | abalone | 4177 | 8 | 359934 | tecator | 240 | 124 |
| 359937 | black_friday | 166821 | 9 | 359939 | topo_2_1 | 8885 | 266 |
| 359950 | boston | 506 | 13 | 359945 | us_crime | 1994 | 126 |
| 359942 | colleges | 7063 | 44 | 359935 | wine_quality | 6497 | 11 |
| 233211 | diamonds | 53940 | 9 | 359940 | yprop_4_1 | 8885 | 251 |
| 359936 | elevators | 16599 | 18 | | | | |

Table 7: Model versions.

| model | benchmarked | latest | package | # configs | compute type |
|---|---|---|---|---|---|
| LightGBM | 3.3.5 | 4.3.0 | lightgbm | 201 | CPU |
| XGBoost | 1.7.6 | 2.0.3 | xgboost | 201 | CPU |
| CatBoost | 1.2.1 | 1.2.3 | catboost | 201 | CPU |
| RandomForest | 1.1.1 | 1.4.1 | scikit-learn | 201 | CPU |
| ExtraTrees | 1.1.1 | 1.4.1 | scikit-learn | 201 | CPU |
| LinearModel | 1.1.1 | 1.4.1 | scikit-learn | 51 | CPU |
| KNeighbors | 1.1.1 | 1.4.1 | scikit-learn | 51 | CPU |
| MLP | 2.0.1 | 2.2.1 | torch | 201 | CPU |
| FTTransformer | 2.0.1 | 2.2.1 | torch | 1 | GPU |
| TabPFN | 0.1.7 | 0.1.10 | tabpfn | 1 | GPU |

### E.1 Model Config Hyperparameters

For a comprehensive list of each config's model hyperparameters for each family, refer to the config files located at the following link: `https://github.com/autogluon/tabrepo/tree/main/data/configs`

To regenerate these files, execute this script: `https://github.com/autogluon/tabrepo/blob/main/scripts/run_generate_all_configs.py`

To see the exact search spaces used, refer to the files located here: `https://github.com/autogluon/tabrepo/tree/main/tabrepo/models`

### E.2 MLP

The MLP used in this work is the one introduced in Erickson et al. (2020) and is implemented in PyTorch. We use a max epochs of 500 and AutoGluon's default early stopping logic.

```
{
    'learning_rate': Real(1e-4, 3e-2, default=3e-4, log=True),
    'weight_decay': Real(1e-12, 0.1, default=1e-6, log=True),
    'dropout_prob': Real(0.0, 0.4, default=0.1),
    'use_batchnorm': Categorical(False, True),
    'num_layers': Int(1, 5, default=2),
    'hidden_size': Int(8, 256, default=128),
    'activation': Categorical('relu', 'elu'),
}
```

### E.3 CatBoost

For CatBoost we use a max iterations of 10000 and AutoGluon's default early stopping logic.

```
{
    'learning_rate': Real(lower=5e-3, upper=0.1, default=0.05, log=True),
    'depth': Int(lower=4, upper=8, default=6),
    'l2_leaf_reg': Real(lower=1, upper=5, default=3),
    'max_ctr_complexity': Int(lower=1, upper=5, default=4),
    'one_hot_max_size': Categorical(2, 3, 5, 10),
    'grow_policy': Categorical("SymmetricTree", "Depthwise"),
}
```

### E.4 LightGBM

For LightGBM we use a max iterations of 10000 and AutoGluon's default early stopping logic.

```
{
    'learning_rate': Real(lower=5e-3, upper=0.1, default=0.05, log=True),
    'feature_fraction': Real(lower=0.4, upper=1.0, default=1.0),
    'min_data_in_leaf': Int(lower=2, upper=60, default=20),
    'num_leaves': Int(lower=16, upper=255, default=31),
    'extra_trees': Categorical(False, True),
}
```

### E.5 XGBoost

For XGBoost we use a max iterations of 10000 and AutoGluon's default early stopping logic.

```
{
    'learning_rate': Real(lower=5e-3, upper=0.1, default=0.1, log=True),
    'max_depth': Int(lower=4, upper=10, default=6),
    'min_child_weight': Real(0.5, 1.5, default=1.0),
    'colsample_bytree': Real(0.5, 1.0, default=1.0),
    'enable_categorical': Categorical(True, False),
}
```

### E.6 Extra-trees

For all Extra Trees models we use 300 trees.

```
{
    'max_leaf_nodes': Int(5000, 50000),
    'min_samples_leaf': Categorical(1, 2, 3, 4, 5, 10, 20, 40, 80),
    'max_features': Categorical('sqrt', 'log2', 0.5, 0.75, 1.0)
}
```

### E.7 Random-forest

For all Random Forest models we use 300 trees.

```
{
    'max_leaf_nodes': Int(5000, 50000),
    'min_samples_leaf': Categorical(1, 2, 3, 4, 5, 10, 20, 40, 80),
    'max_features': Categorical('sqrt', 'log2', 0.5, 0.75, 1.0)
}
```

### E.8 Linear

```
{
    "C": Real(lower=0.1, upper=1e3, default=1),
    "proc.skew_threshold": Categorical(0.99, 0.9, 0.999, None),
    "proc.impute_strategy": Categorical("median", "mean"),
    "penalty": Categorical("L2", "L1"),
}
```

### E.9 KNN

```
{
    'n_neighbors': Categorical(3, 4, 5, 6, 7, 8, 9, 10, 11, 13, 15, 20, 30, 40, 50)
    ,
    'weights': Categorical('uniform', 'distance'),
    'p': Categorical(2, 1),
}
```

### E.10 TabPFN

We use the default setting for TabPFN and did not run additional configs. We tested specifying N_ensemble_configurations to values above 1 which saw marginal improvement, but decided to keep only the default to simplify comparisons. TabPFN is intended for datasets with at most 1000 rows of training data, whereas TabRepo's dataset suite includes many datasets with far more than 1000 rows. Because of this, we emphasize that our findings are not representative of TabPFN's ideal usage scenario, but we include TabPFN to improve the comprehensiveness of our work.

### E.11 FT-Transformer

We use the default setting for FT-Transformer and did not run additional configs. We use the AutoGluon implementation of FT-Transformer, which may have slight differences from the original implementation. We ran FT-Transformer on a GPU instance to enable it to train on more datasets to completion. Because we used a GPU instance, our Portfolio reported in the main text does not include FT-Transformer. We did not run additional configs due to cost reasons, as GPU instances are significantly more expensive than CPU instances.

The default FT-Transformer hyperparameters are noted below.

```
1  {
2      "model.ft_transformer.n_blocks": 3,
3      "model.ft_transformer.d_token": 192,
4      "model.ft_transformer.adapter_output_feature": 192,
5      "model.ft_transformer.ffn_d_hidden": 192,
6      "optimization.learning_rate": 1e-4,
7      "optimization.weight_decay": 1e-5,
8  }
```

## F  AutoML Framework Details

For each AutoML framework we attempted to use the latest available versions where possible. The precise versions used for each framework are documented in Tab. 8. All AutoML frameworks were ran with the configurations as in Gijsbers et al. (2024). For FLAML, version 2.0 released after we had ran the experiments.

### F.1  Excluded Frameworks

**MLJAR and GAMA.** We did not include the AutoML frameworks MLJAR (Płońska and Płoński, 2021) and GAMA (Gijsbers and Vanschoren, 2021) in the main results due to these frameworks having a significant number of implementation errors when running on the main TabRepo datasets in comparison to the other AutoML systems which had 0 failures. We did include them in Tab. 2 as we directly re-used the 2023 results from Gijsbers et al. (2024).

**NaiveAutoML.** NaiveAutoML (Mohr and Wever, 2022) is a recently introduced AutoML system that we did not include in TabRepo. This is because we ran AutoMLBenchmark using identical configurations to Gijsbers et al. (2024), which at the time had mentioned suboptimal settings related to NaiveAutoML's "EXECUTION_TIMEOUT" parameter which they detail in their paper leading to poor results. In order to avoid misrepresenting NaiveAutoML's performance, we decided to exclude it due to the configuration issues. After running our experiments, we later learned from the NaiveAutoML authors that the issues within AMLB had been resolved, but due to a lack of compute budget to re-run experiments NaiveAutoML had not been re-run with the fixed version in Gijsbers et al. (2024). We will consider adding NaiveAutoML to the benchmarked frameworks in future iterations of TabRepo with the author's recommended configuration.

**Auto-WEKA.** As done in Gijsbers et al. (2024), we exclude Auto-WEKA (Thornton et al., 2013) due to its performance and lack of updates in prior iterations of AMLB.

**TPOT.** We exclude TPOT (Olson and Moore, 2019) due to performance and stability issues, which led it to having a similar amount of failures as MLJAR and GAMA in Gijsbers et al. (2024).

### F.2  AutoGluon

We run AutoGluon with the "best_quality" preset for all comparisons, as done in Gijsbers et al. (2024). This setting is used to achieve the strongest predictive quality.

In Tab. 2, we use AutoGluon 1.0 for comparisons. The learned portfolio used in AutoGluon 1.0 is Portfolio with size 100 as opposed to the Portfolio with size 200 used for comparison in Tab. 1. The size 100 Portfolio is slightly worse in accuracy, but achieves faster latency, faster training time, and lower disk space usage, which is why it was chosen as the default for AutoGluon 1.0. We use an m5.2xlarge instance for training AutoGluon on AMLB to replicate the compute used by other systems.

Table 8: AutoML framework versions.

| framework | benchmarked | latest | package |
|---|---|---|---|
| AutoGluon | 0.8.2 | 1.0.0 | autogluon |
| auto-sklearn | 0.15.0 | 0.15.0 | auto-sklearn |
| auto-sklearn 2 | 0.15.0 | 0.15.0 | auto-sklearn |
| FLAML | 1.2.4 | 2.1.1 | flaml |
| H2O AutoML | 3.40.0.4 | 3.44.0.3 | h2o |
| LightAutoML | 0.3.7.3 | 0.3.8.1 | lightautoml |

### F.3 Auto-Sklearn 2

**Meta-Learning**. Auto-Sklearn 2 uses meta-learning to improve the quality of its results. Since the datasets used to train its meta-learning algorithm are present in TabRepo, the performance of Auto-Sklearn 2 may be overly optimistic as it may be choosing to train model hyperparameters known to achieve strong test scores on a given dataset. This issue is detailed in section 5.3.3 of Gijsbers et al. (2024). Following Gijsbers et al. (2024), we ultimately decide to keep Auto-Sklearn 2's results as a useful comparison point.

**Regression**. Auto-Sklearn 2 is incompatible with regression tasks. For regression tasks, we use the result from Auto-Sklearn 1.

## G Studying Portfolio Model Prioritization

Fig. 6 shows the Portfolio composition by model family averaged across all 200 leave-one-dataset-out evaluations. Interestingly, as can be seen in the top figure, the first 4 model family selections are identical across all leave-one-out evaluations.

CatBoost is picked first as it is the strongest individual model. The second pick is an MLP model, which is expected, as tree models are known to ensemble well with neural networks, which was observed in Fig 1a. The third pick is LightGBM. Referencing back to Fig. 5c, we observed that a Portfolio of size 3 is sufficient to outperform all AutoML methods besides AutoGluon. With this visualization, we now know that this size 3 portfolio is very consistently structured, and simply sequentially trains a CatBoost, MLP, and LightGBM model, followed by a simple weighted ensemble.

Going further, the fourth pick is CatBoost, the first model family repeat. The fifth pick shows the first instance where some of the leave-one-out Portfolios begin to select different families at the same Portfolio position. While roughly 90% pick LightGBM, the remaining 10% pick KNN. We see that the probabilities reverse with the 6th position, likely meaning that the two diverging groups have now converged back to the same overall composition, with slightly different pick orders.

For the 7th position, we see MLP picked a second time, followed by the first XGBoost pick in the 8th position, and the first ExtraTrees pick in the 9th position. CatBoost is picked for the 3rd and 4th time in the 10th and 11th positions. The last remaining unpicked families, Linear models and Random Forest, are picked for the first time at positions 12 and 13.

Beyond position 13, the leave-one-out Portfolios have diverged sufficiently that there is no more unanimous picks at a given position. However, looking at the bottom figure, we see that the overall proportion of each family's inclusion in the Portfolio begins to stabilize. We observe that MLP becomes the most frequently picked family, followed by LightGBM and CatBoost.

Given that all 8 model families were picked at least once by position 13 shows that each family contributes meaningfully to the performance of the Portfolio, and no model family is irrelevant. KNN, which is by far the worst performing model in isolation, was the 4th model family picked by the portfolio, even ahead of the much stronger XGBoost. We suspect that this is due to the large

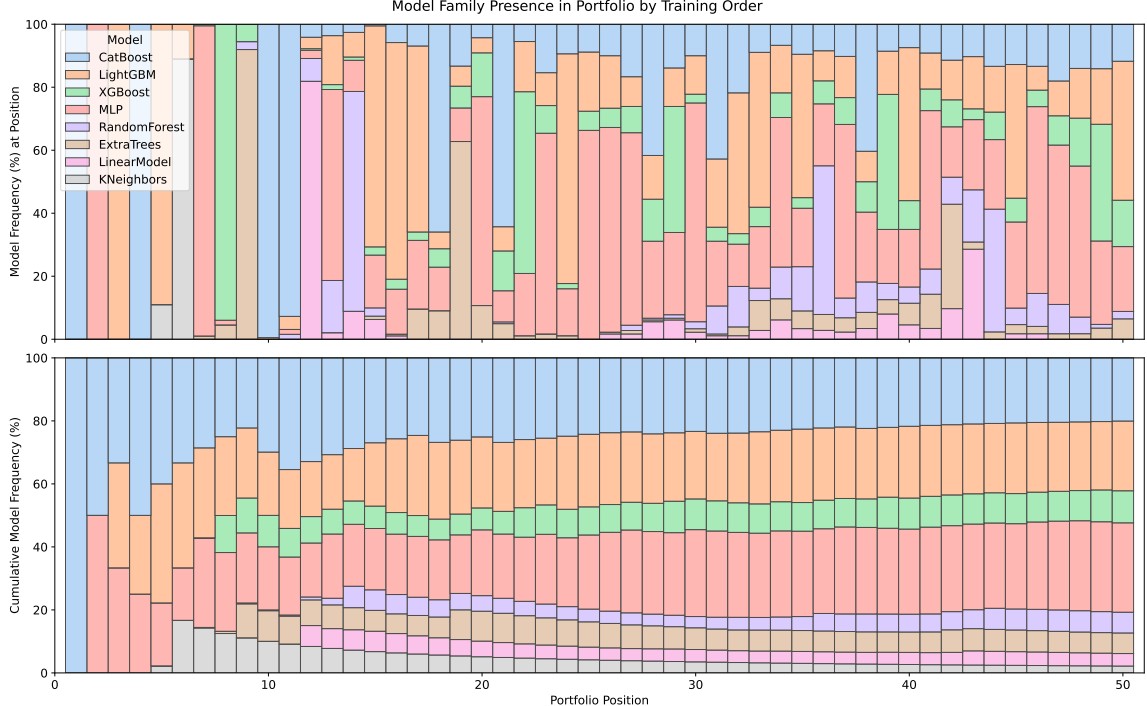

Figure 6: Visualization of the Portfolio composition and model selection order, averaged across all 200 leave-one-dataset-out evaluations. The top figure shows the percentage of the time each model family is picked at a given position N in the Portfolio. The bottom figure shows the cumulative percentage each model family is used in the Portfolio in the first N Portfolio positions.

Table 9: Models exceeding 2800 seconds training time per family.

| Method | # Exceeding | % Exceeding |
|--------|-------------|-------------|
| CatBoost | 11851 | 9.8% |
| XGBoost | 2532 | 2.1% |
| LightGBM | 1025 | 0.8% |
| LinearModel | 69 | 0.2% |
| FTTransformer | 40 | 7.5% |
| MLP | 14 | 0.0% |

diversity contributed by KNN to the Portfolio, whereas XGBoost has to compete with CatBoost and LightGBM, which share very similar architectures and performance characteristics.

The current logic that picks the Portfolio composition is blind to model training times. We expect significant improvements can be made to the strength of the portfolio model order by incentivizing the Portfolio to pick cheaper models with higher priority, improving the any-time performance of the Portfolio. We leave further investigations to future work.

## H  Impact of Time Limit

Here we analyze the impact of the one hour time limit used to avoid overly long training times for models. The number of models nearing or reaching the time limit (>=2800s) is 15531 (out of 782060), or 1.99% of models.

Table 10: Models exceeding 2800 seconds training time per dataset.

| Task ID | Dataset | # Exceeding | % Exceeding |
|---|---|---|---|
| 359986 | robert | 1797 | 45.8% |
| 168909 | dilbert | 1389 | 35.4% |
| 3481 | isolet | 1046 | 26.7% |
| 168300 | UMIST_Faces_Cropped | 834 | 21.3% |
| 359985 | volkert | 802 | 20.4% |
| 233212 | Allstate_Claims_Seve | 667 | 17.0% |
| 14970 | har | 619 | 15.8% |
| 3980 | OVA_Prostate | 611 | 15.6% |
| 3995 | OVA_Colon | 570 | 14.5% |
| 317614 | Yolanda | 558 | 14.2% |
| 359953 | micro-mass | 548 | 14.0% |
| 3968 | OVA_Kidney | 510 | 13.0% |
| 4000 | OVA_Ovary | 460 | 11.7% |
| 211986 | Diabetes130US | 459 | 11.7% |
| 3964 | OVA_Lung | 456 | 11.6% |
| 9972 | kr-vs-k | 450 | 11.5% |
| 2076 | kropt | 450 | 11.9% |
| 360859 | Indian_pines | 420 | 10.7% |
| 168910 | fabert | 356 | 9.1% |
| 3976 | OVA_Endometrium | 334 | 8.5% |
| 41 | soybean | 290 | 7.7% |
| 359961 | mfeat-factors | 284 | 7.2% |
| 361330 | Traffic_violations | 264 | 6.7% |
| 9964 | semeion | 249 | 6.6% |
| 359977 | connect-4 | 190 | 5.0% |
| 359993 | okcupid-stem | 173 | 4.4% |
| 211979 | jannis | 134 | 3.4% |
| 189922 | gina | 129 | 3.3% |
| 146800 | MiceProtein | 127 | 3.2% |
| 190412 | arcene | 120 | 3.1% |
| 125922 | texture | 63 | 1.6% |
| 3512 | synthetic_control | 50 | 1.3% |
| 45 | splice | 39 | 1.0% |
| 361331 | GAMETES_Epistasis_2- | 24 | 0.6% |
| 359980 | nomao | 18 | 0.5% |
| 359991 | kick | 13 | 0.3% |
| 146802 | collins | 7 | 0.2% |
| 359990 | MiniBooNE | 5 | 0.1% |
| 359981 | jungle_chess_2pcs_ra | 3 | 0.1% |
| 359937 | black_friday | 3 | 0.1% |
| 9979 | cardiotocography | 3 | 0.1% |
| 219 | electricity | 3 | 0.1% |
| 168868 | APSFailure | 3 | 0.1% |
| 9976 | madelon | 1 | 0.0% |

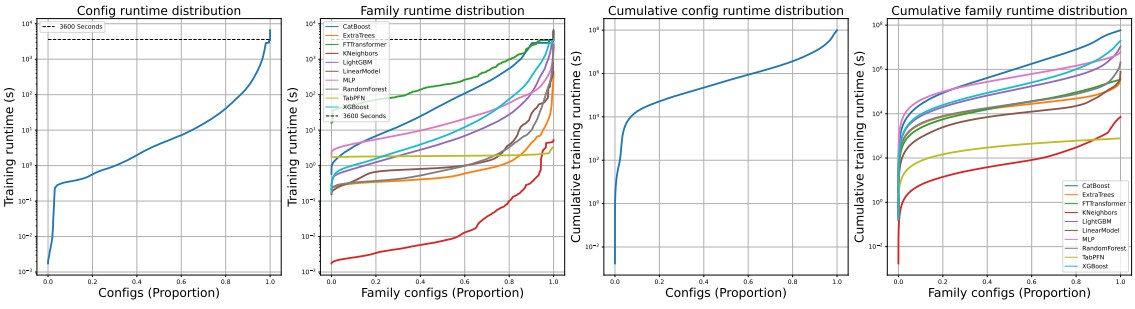

Figure 7: Training time distribution across all datasets for (a) all configurations and (b) grouped by family, as well as the cumulative total training time across all datasets for (c) all configurations and (d) grouped by family.

We can see in Tab. 10 that the majority of cases where the time limit is reached is on large datasets with many features and many classes. For example, the robert dataset has 7200 features and 10 classes, leading to long training times for tree methods that train one-vs-all in multiclass classification. The same can be said for dilbert (2000 features, 5 classes), isolet (617 features, 26 classes), and UMIST_Faces_Cropped (10304 features, 20 classes). The families that are early stopped are almost exclusively GBMs, particularly CatBoost, as highlighted in Tab. 9. FT-Transformer, even with GPU, is early stopped in 7.5% of cases, reflecting its expensive compute requirement compared to other methods.

We see from Fig. 7c that the 10% most expensive models contribute 90% of the overall runtime cost, and the 40% most expensive models contribute 99% of the overall runtime cost. In Fig. 7d, we see that CatBoost's overall runtime is larger than all other families combined. Note that LinearModel slightly exceeded 3600 seconds in a few cases due to the early stopping implementation.

Given the slope of Fig. 7a prior to early stopping triggering, we expect that fully training the 2% most expensive models would over double the compute cost of the entire benchmark. Meanwhile, the most heavily impacted method, CatBoost, is still comfortably the best performing method in TabRepo, and therefore we expect that the one hour time limit does not alter our overall conclusions nor diminish the utility TabRepo provides to researchers. For future work that would tackle larger datasets a time limit greater than 1 hour, larger CPU core counts, and/or GPU acceleration should be considered.

# I Portfolio results when including models requiring GPU

Here, we report results when also including models that requires a GPU. In addition to previous models, we also consider:

- TabPFN (Hollmann et al., 2022) which is transformer model for tabular data pretrained on a collection of artificial datasets that performs attention over rows

- FT-Transformer (Gorishniy et al., 2021) which is a transformer trained on a dataset at hand and performs attention over columns

For TabPFN and FT-Transformer, we measure results on a g4.2xlarge instance. We run only the default configuration for FT-Transformer due to the large training cost to obtain results on all tasks on a GPU machine, we also ran a single configuration for TabPFN.

We report those results separately because those models requires an additional GPU as opposed to the models presented in the main sections which pose different hardware constraint cost.

Some of the models fail because of algorithm errors (for instance TabPFN only supports 100 features currently) or hardware errors (out-of-memory errors). In case of failure, we impute the model predictions with the baseline used when portfolio configuration times out (e.g. the default configuration of Extra-trees), this baseline always take less than 5 seconds to run.

As one can see in Tab. 14, FT-Transformer performs in-between MLPs and the best boosted tree methods. Regarding TabPFN, the method does not reach the performance of top methods yet which is due to high failure rates due to current method limitations on large datasets[6] and also due to the method not being able to currently effectively exploit large number of rows.

The results of portfolio improves slightly given the additional model diversity which can be seen by looking at Tab. 13 which reports the win-rate against AutoML baselines. In particular, the win rate is improved from 56.8% to 57.5%. We expect with additional configs for these model families that the performance of the portfolio would improve further.

---

[6]The failure rate is ≈ 30% as the method only supports 100 features and 10 classes.

Table 11: Win rate comparison for 4 hour time limit with the same methodology as Erickson et al. (2020). Win rate is computed against a portfolio ensemble (ties count as 0.5). The re-scaled loss is calculated by setting the best solution to 0 and the worst solution to 1 on each dataset, and then normalizing and taking the mean across all datasets. Rank, fit time, and infer time are averaged over all tasks.

| method | winrate | > | < | = | time fit (s) | time infer (s) | loss (rescaled) | rank |
|---|---|---|---|---|---|---|---|---|
| **Portfolio (ensemble) (ours) (4h)** | **0.500** | 0 | 0 | 200 | 6275.5 | 0.050 | **0.233** | **3.042** |
| AutoGluon (4h) | 0.432 | 84 | 111 | 5 | 5583.1 | 0.062 | 0.288 | 3.435 |
| Autosklearn2 (4h) | 0.330 | 65 | 133 | 2 | 14415.9 | 0.013 | 0.403 | 4.380 |
| Lightautoml (4h) | 0.270 | 52 | 144 | 4 | 9173.9 | 0.298 | 0.433 | 4.643 |
| CatBoost (tuned + ensemble) (4h) | 0.242 | 47 | 150 | 3 | 9065.2 | 0.012 | 0.508 | 5.010 |
| Autosklearn (4h) | 0.280 | 54 | 142 | 4 | 14413.6 | 0.009 | 0.516 | 5.055 |
| Flaml (4h) | 0.263 | 50 | 145 | 5 | 14267.0 | 0.002 | 0.535 | 5.122 |
| H2oautoml (4h) | 0.225 | 42 | 152 | 6 | 13920.1 | 0.002 | 0.562 | 5.312 |

Table 12: Win rate comparison for 1 hour time limit with the same approach used as for Tab. 11.

| method | winrate | > | < | = | time fit (s) | time infer (s) | loss (rescaled) | rank |
|---|---|---|---|---|---|---|---|---|
| **Portfolio (ensemble) (ours) (1h)** | **0.500** | 0 | 0 | 200 | 2318.2 | 0.022 | **0.230** | **3.145** |
| AutoGluon (1h) | 0.453 | 88 | 107 | 5 | 2283.7 | 0.033 | 0.273 | 3.310 |
| Autosklearn2 (1h) | 0.383 | 74 | 121 | 5 | 3611.2 | 0.010 | 0.393 | 4.272 |
| Lightautoml (1h) | 0.292 | 56 | 139 | 5 | 3002.7 | 0.099 | 0.408 | 4.435 |
| CatBoost (tuned + ensemble) (1h) | 0.247 | 48 | 149 | 3 | 2923.3 | 0.005 | 0.531 | 5.140 |
| Autosklearn (1h) | 0.305 | 60 | 138 | 2 | 3612.0 | 0.007 | 0.536 | 5.173 |
| H2oautoml (1h) | 0.203 | 39 | 158 | 3 | 3572.8 | 0.002 | 0.506 | 5.195 |
| Flaml (1h) | 0.263 | 50 | 145 | 5 | 3623.8 | 0.001 | 0.552 | 5.330 |

## J Additional results Portfolios

### J.1 Portfolio win-rate comparison

We calculate win-rates, re-scaled loss, and average ranks between the Portfolio and the AutoML systems in Tab. 12 and Tab. 11 for 1 and 4 hour time limits respectively with the same evaluation protocol as Erickson et al. (2020). In both cases, Portfolio achieves the best win-rate, re-scaled loss, and average rank across all methods at the given time constraint.

### J.2 Performance on lower fitting budgets

In section 5, we reported results for 1h, 4h fitting budgets which are standard settings (Erickson et al., 2020; Gijsbers et al., 2024). Given space constraint, we only showed the full table for 4h results in the main, the results for 1h results is shown in Tab. 15. Here the anytime portfolio strategy matches AutoGluon on normalized error and outperforms it on rank while having around 30% lower latency.

Table 13: Win rate comparison with additional models requiring a GPU defined in Section I for 4 hour time limit.

| method | winrate | > | < | = | time fit (s) | time infer (s) | loss (rescaled) | rank |
|---|---|---|---|---|---|---|---|---|
| Portfolio-GPU (ensemble) | 0.500 | 0 | 0 | 200 | 6597.5 | 0.061 | 0.239 | 3.115 |
| AutoGluon best | 0.425 | 80 | 110 | 10 | 5583.1 | 0.062 | 0.290 | 3.442 |
| Autosklearn2 | 0.350 | 68 | 128 | 4 | 14415.9 | 0.013 | 0.404 | 4.360 |
| Lightautoml | 0.287 | 56 | 141 | 3 | 9173.9 | 0.298 | 0.434 | 4.625 |
| CatBoost (tuned + ensemble) | 0.245 | 47 | 149 | 4 | 9065.2 | 0.012 | 0.506 | 5.008 |
| Autosklearn | 0.290 | 56 | 140 | 4 | 14413.6 | 0.009 | 0.515 | 5.045 |
| Flaml | 0.295 | 56 | 138 | 6 | 14267.0 | 0.002 | 0.533 | 5.090 |
| H2oautoml | 0.223 | 41 | 152 | 7 | 13920.1 | 0.002 | 0.565 | 5.315 |

Table 14: Results with additional models defined in Section I. Normalized error, rank, training and inference time are averaged over all tasks given 4h training budget.

| method | normalized-error | rank | time fit (s) | time infer (s) |
|---|---|---|---|---|
| Portfolio-GPU (ensemble) | 0.362 | 174.6 | 6597.5 | 0.061 |
| Portfolio (ensemble) | 0.365 | 168.7 | 6275.5 | 0.050 |
| AutoGluon best | 0.389 | 208.2 | 5583.1 | 0.062 |
| Portfolio | 0.434 | 232.5 | 6275.5 | 0.012 |
| Portfolio-GPU | 0.437 | 236.6 | 6597.5 | 0.013 |
| Autosklearn2 | 0.455 | 243.5 | 14415.9 | 0.013 |
| Lightautoml | 0.466 | 246.1 | 9173.9 | 0.298 |
| Flaml | 0.513 | 317.8 | 14267.0 | 0.002 |
| CatBoost (tuned + ensemble) | 0.524 | 267.3 | 9065.2 | 0.012 |
| H2oautoml | 0.526 | 337.0 | 13920.1 | 0.002 |
| CatBoost (tuned) | 0.534 | 284.7 | 9065.2 | 0.002 |
| LightGBM (tuned + ensemble) | 0.534 | 268.7 | 3528.9 | 0.010 |
| LightGBM (tuned) | 0.566 | 304.2 | 3528.9 | 0.001 |
| CatBoost (default) | 0.586 | 341.2 | 456.8 | 0.002 |
| MLP (tuned + ensemble) | 0.594 | 402.5 | 5771.8 | 0.098 |
| XGBoost (tuned + ensemble) | 0.628 | 357.9 | 4972.7 | 0.013 |
| MLP (tuned) | 0.634 | 451.9 | 5771.8 | 0.014 |
| XGBoost (tuned) | 0.638 | 376.5 | 4972.7 | 0.002 |
| FTTransformer (default) | 0.690 | 532.1 | 567.4 | 0.003 |
| LightGBM (default) | 0.714 | 491.5 | 55.7 | 0.001 |
| XGBoost (default) | 0.734 | 522.2 | 75.1 | 0.002 |
| MLP (default) | 0.772 | 629.4 | 38.2 | 0.015 |
| ExtraTrees (tuned + ensemble) | 0.782 | 544.2 | 538.3 | 0.001 |
| ExtraTrees (tuned) | 0.802 | 572.5 | 538.3 | 0.000 |
| RandomForest (tuned + ensemble) | 0.803 | 578.3 | 1512.2 | 0.001 |
| RandomForest (tuned) | 0.816 | 598.0 | 1512.2 | 0.000 |
| TabPFN (default) | 0.837 | 731.9 | 3.8 | 0.016 |
| LinearModel (tuned + ensemble) | 0.855 | 873.8 | 612.4 | 0.038 |
| LinearModel (tuned) | 0.862 | 891.6 | 612.4 | 0.006 |
| ExtraTrees (default) | 0.883 | 788.6 | 3.0 | 0.000 |
| RandomForest (default) | 0.887 | 773.9 | 13.8 | 0.000 |
| LinearModel (default) | 0.899 | 940.1 | 7.1 | 0.014 |
| KNeighbors (tuned + ensemble) | 0.928 | 980.8 | 12.0 | 0.001 |
| KNeighbors (tuned) | 0.937 | 1016.5 | 12.0 | 0.000 |
| KNeighbors (default) | 0.973 | 1149.1 | 0.6 | 0.000 |

Table 15: Normalized error, rank, training and inference time averaged over all tasks given 1h training budget.

| method | normalized-error | rank | time fit (s) | time infer (s) |
|---|---|---|---|---|
| Portfolio (ensemble) | 0.376 | 176.0 | 2318.2 | 0.022 |
| AutoGluon | 0.393 | 206.7 | 2283.7 | 0.033 |
| Portfolio | 0.444 | 242.3 | 2318.2 | 0.012 |
| Autosklearn2 | 0.470 | 257.1 | 3611.2 | 0.010 |
| Lightautoml | 0.479 | 252.3 | 3002.7 | 0.099 |
| H2oautoml | 0.524 | 325.3 | 3572.8 | 0.002 |
| Flaml | 0.530 | 340.3 | 3623.8 | 0.001 |
| LightGBM (tuned + ensemble) | 0.537 | 271.8 | 1643.3 | 0.008 |
| LightGBM (tuned) | 0.570 | 307.2 | 1643.3 | 0.002 |
| CatBoost (tuned + ensemble) | 0.574 | 318.1 | 2923.3 | 0.005 |
| CatBoost (tuned) | 0.580 | 333.2 | 2923.3 | 0.002 |
| CatBoost (default) | 0.586 | 341.2 | 456.8 | 0.002 |
| MLP (tuned + ensemble) | 0.600 | 407.9 | 2559.8 | 0.120 |
| XGBoost (tuned + ensemble) | 0.638 | 371.6 | 1864.5 | 0.013 |
| MLP (tuned) | 0.638 | 454.4 | 2559.8 | 0.014 |
| XGBoost (tuned) | 0.648 | 390.2 | 1864.5 | 0.002 |
| FTTransformer (default) | 0.690 | 532.1 | 567.4 | 0.003 |
| LightGBM (default) | 0.714 | 491.5 | 55.7 | 0.001 |
| XGBoost (default) | 0.734 | 522.2 | 75.1 | 0.002 |
| MLP (default) | 0.772 | 629.4 | 38.2 | 0.015 |
| ExtraTrees (tuned + ensemble) | 0.782 | 544.3 | 382.6 | 0.001 |
| ExtraTrees (tuned) | 0.802 | 572.7 | 382.6 | 0.000 |
| RandomForest (tuned + ensemble) | 0.803 | 579.8 | 668.7 | 0.001 |
| RandomForest (tuned) | 0.815 | 597.5 | 668.7 | 0.000 |
| TabPFN (default) | 0.837 | 731.9 | 3.8 | 0.016 |
| LinearModel (tuned + ensemble) | 0.855 | 874.4 | 374.3 | 0.038 |
| LinearModel (tuned) | 0.862 | 892.5 | 374.3 | 0.006 |
| ExtraTrees (default) | 0.883 | 788.6 | 3.0 | 0.000 |
| RandomForest (default) | 0.887 | 773.9 | 13.8 | 0.000 |
| LinearModel (default) | 0.899 | 940.1 | 7.1 | 0.014 |
| KNeighbors (tuned + ensemble) | 0.928 | 980.8 | 12.0 | 0.001 |
| KNeighbors (tuned) | 0.937 | 1016.5 | 12.0 | 0.000 |
| KNeighbors (default) | 0.973 | 1149.1 | 0.6 | 0.000 |

