# OpenReview forum: "TabRepo: A Large Scale Repository of Tabular Model Evaluations and its AutoML Applications"
_automl.cc/AutoML/2024/ABCD_Track — AutoML 2024 (ABCD Track)_

### Official Review · Reviewer_x7jZ · 2024-03-24

**Potential Impact On The Field Of Automl Rating:** 4
**Technical Quality And Correctness Rating:** 4
**Clarity:** The paper is well written and is easy…
**Clarity Rating:** 4

**Summary Of Contributions:**

The paper publishes a large data set of evaluations of models trained on tabular data, which can be used for benchmarking hyperparameter tuning and model section algorithms.
In addition to its comprehensive scale, the data set also includes predictions made by the include models when trained on particular training sets; this allows for rapid calculations of predictions made by ensembles of models without requiring retraining.
The authors also conduct experiments showcasing the extensiveness of the data set and demonstrating the ability to use the data set to perform transfer learning.

**Actions Required To Increase Overall Recommendation:**

A clarification on whether the training duration of the models include in the data set is sufficient will be appreciated.

**Overall Review:**

The paper makes an important contribution to the field of AutoML.
I expect that the published data set will be extensively used by subsequent work on hyperparameter tuning and ensemble models.
I have a minor concern about how long the models are trained when building the data set (see above).

**Potential Impact On The Field Of Automl:**

The paper's potential impact is substantial.
I would likely cite the paper.

**Review Confidence:**

3

**Review Rating:**

8

**Review Summary:**

The paper makes important contributions that can spark many following works.

**Technical Quality And Correctness:**

The paper's approach and experiments are of high quality.
The authors successfully demonstrate the importance of the data set and its various uses.
The only complaint I have is about the fact that for the models included in the data set, all training is stopped after one hour, which seems low to me, but perhaps that is sufficient for (small) tabular data.
More discussion on this point could prove beneficial.

---

### Official Review · Reviewer_SSjc · 2024-03-27

**Potential Impact On The Field Of Automl Rating:** 3
**Technical Quality And Correctness Rating:** 4
**Clarity Rating:** 3

**Summary Of Contributions:**

This paper presents TabRepo, a new dataset comprising evaluations and predictions from more than 1000 models across 200 regression and classification datasets. This dataset facilitates various analyses, including comparing Hyperparameter Optimization with existing AutoML systems and incorporating ensembling effortlessly using precomputed model predictions. Additionally, the authors show that TabRepo enables transfer learning, where standard techniques surpass current top-performing tabular systems across many metrics.

--Post rebuttal--

I have no further concerns and I thank the authors for responding to my feedback. I will keep my recommendation to accept.

**Actions Required To Increase Overall Recommendation:**

I'd request some comments on the points on room for improvement I mentioned, but I think this is a solid work and already merits acceptance.

**Clarity:**

The paper is written clearly. My only concern on the illustrations is that the font size of the figures are small and it'd be beneficial to make the font at least as large as the one used in the rest of the paper.

**Overall Review:**

Overall, I think this paper presents a useful dataset resource that is likely to be quite influential in the area of tabular learning as I believe it can significantly lower the research barrier for many time-consuming tasks that are otherwise inaccessible to many researchers. The evaluation is thorough, and the dataset has good coverage in terms of diversity in task, model and configurations and I commend the authors for their efforts in curating such a dataset. Some possible areas of improvements include:

- Given the recent research interest in using LLMs for tabular tasks, I wonder is it possible also to add some kind of LLM evaluation as a method in the paper? I believe this also creates a unique opportunity for the authors to compare and contrast LLM-based approaches against the more classical machine learning approaches presented in the paper on equal ground. Insights from this could also be very useful for the community.

**Potential Impact On The Field Of Automl:**

Learning on structured data is a very important research direction both within AutoML and across broader AI community. The dataset introduced in this paper is large-scale, covers diverse task and model configurations, and is likely to democratize research that was previously unaccessible to broader audience. As such, I think this paper is likely to have a medium-to-high potential impact on AutoML (I'd rate it between 3 and 4 in the impact rating below).

**Review Confidence:**

3

**Review Rating:**

8

**Review Summary:**

Please see overall review.

**Technical Quality And Correctness:**

This is a benchmark paper, and the authors mainly evaluated existing models on the curated datasets, and thus I do not see any soundness issues. The only concern is that 3 seeds might not be enough to conclude statistical significance. Having said that, I do acknowledge the extensive amount of computation required to produce this dataset and it's easier said than done to create more seeds, and using 3 seeds is also on par existing popular AutoML benchmarks like NAS-Bench-201, so I will not hold this as a weakness against the authors, although I'd encourage them to discuss and acknowledge potential concerns here.

---

### Official Review · Reviewer_S4Yo · 2024-03-27

**Potential Impact On The Field Of Automl Rating:** 4
**Technical Quality And Correctness Rating:** 4
**Clarity Rating:** 4

**Summary Of Contributions:**

The paper introduces TabRepo, a large dataset of tabular model evaluations and predictions. By collecting model predictions across seeds and hyperparameter settings, TabRepo allows for quick access to trained models for benchmarking. Through the access to model predictions, users may consider model ensabmling and transfer learning across datasets with low effort, saving on compute and time.

**Actions Required To Increase Overall Recommendation:**

I would be willing to raise my score if additional empirical insights into the dataset were added.

**Clarity:**

The paper is clear in all sections. Given that the core idea is fairly straight-forward, I do not see a major issue. However, the paper would read better if the magnitude (compute resources, number of seeds and settings) were downplayed, perhaps by moving some of it to the appendix.

**Overall Review:**

Very good, extensive work but with some room for improvement. The dataset is substantial, useful and appears simple to use. Moreover, it enables a low-compute options for ensembling and transfer learning across tabular datasets.

Unfortunately, there is some lacking empirical insights into the dataset itself, and the magnitude of the dataset is partly overplayed in the paper to the detriment of more interesting insights.

**Potential Impact On The Field Of Automl:**

High. The paper introduces a new type of dataset/benchmarking suite, which appears relevant, thought-out and convenient.

**Reproducibility:**

No comment.

**Review Confidence:**

4

**Review Rating:**

9

**Review Summary:**

To not repeat myself, I refer the authors to the comments above.

**Technical Quality And Correctness:**

The dataset has obvious appeal. I am not aware of a collection of ready-trained model predictions, which is convenient. It also appears easy-to-use, given that it is structured and constitutes a single look-up.

The paper primarily delves into the scope of the dataset that has been developed. While the work is undoubtedly structured and extensive, the paper overly communicates this aspect to the detriment of interesting insights into the dataset itself. Fig. 2 is a very good add, and it would be good to see more of those. For example:
- For what models do we see substantial variation in performance across seeds/HP settings?
- Which models were particularly time-sensitive?

I do not expect the authors to add these analyses, but I do believe more such figures and insights would strengthen the paper substantially.

---

### Official Review · Reviewer_VV1G · 2024-03-28

**Potential Impact On The Field Of Automl:** I believe the work will have a medium…
**Potential Impact On The Field Of Automl Rating:** 3
**Technical Quality And Correctness Rating:** 3
**Clarity:** The paper is written nicely and it is…
**Clarity Rating:** 4

**Summary Of Contributions:**

The authors propose TabRepo, a tabular benchmark that investigates the performance of different tabular models on a wide range of datasets providing insights on the performance of individual models with default hyperparameter configurations/tuned hyperparameter configurations against one another and against different AutoML frameworks. The authors additionally consider ensembles of the different algorithms from the same family and build a portfolio with the given data providing valuable insights.

**Actions Required To Increase Overall Recommendation:**

- The authors cannot address using random search and not a model-based HPO method as that would require rerunning the whole benchmark.  However, the authors can add the ResNet architecture. It would require 200 X 3 X 8 X D runs.
- It would have been interesting to see an investigation of how the hyperparameters of every method affect performance and what the most important hyperparameters are.

**Overall Review:**

- **Strengths**
    - Extensive number of datasets considered for both classification and regression.
    - Results provided for both default hyperparameters and tuned hyperparameters of single models, additionally for ensembles.
    - Various ablations for the portfolio generation.
    - The work provides a valuable benchmark for the community to use in the future.
    - The work provides model predictions and not only the metric performance.


- **Weaknesses:**
    - To not become repetitive see "Technical Quality And Correctness".
    - It would have been interesting to see an investigation of how different hyperparameters affect performance for the different methods.

- **Typos:**
    - The number of models mentioned in the abstract of the submission with the number of models in the manuscript of the submission differ.

**Review Confidence:**

5

**Review Rating:**

8

**Review Summary:**

- See above.

**Technical Quality And Correctness:**

I believe the authors have done an extensive comparison. What I would have preferred to be in the provided results would be:

- The tabular ResNet architecture [1][2] as in [3].
- Not using random search for hyperparameter optimization but instead a model-based method. Random search will tend to benefit more methods that have a robust performance despite the used hyperparameters, such as CatBoost, XGBoost, etc, which can, in turn, provide an unfair comparison. A more guided search would be more beneficial.

[1] Gorishniy, Y., Rubachev, I., Khrulkov, V., & Babenko, A. (2021). Revisiting deep learning models for tabular data. Advances in Neural Information Processing Systems, 34, 18932-18943.

[2] Kadra, A., Lindauer, M., Hutter, F., & Grabocka, J. (2021). Well-tuned simple nets excel on tabular datasets. Advances in neural information processing systems, 34, 23928-23941.

[3] McElfresh, D., Khandagale, S., Valverde, J., Prasad C, V., Ramakrishnan, G., Goldblum, M., & White, C. (2024). When do neural nets outperform boosted trees on tabular data?. Advances in Neural Information Processing Systems, 36.

---

### Meta-Review · Area_Chair_GkjB · 2024-04-22

**Paper Recommendation:** Accept
**Confidence:** 5

**Metareview:**

The reviewers unanimously recommend to accept the paper. They agree that the presented repository is useful and likely to have a medium to large impact on the community, but it will certainly be useful.

---

### Decision · Program_Chairs · 2024-04-29

**Decision:**

Accept

**Comment:**

Thank you for submitting your paper. We are happy to tell you that we accept your paper to the main track. See you in Paris.